# REAR: RETRIEVAL-AUGMENTED EGOCENTRIC ACTION RECOGNITION

## ABSTRACT

Egocentric Action Recognition (EAR) aims to identify fine-grained actions and interacted objects from first-person videos, forming a core task in egocentric video understanding. Despite recent progress, EAR remains challenged by limited data scale, annotation quality, and long-tailed class distributions. To address these issues, we propose REAR, a Retrieval-augmented framework for EAR that leverages external third-person (exocentric) videos as auxiliary knowledge—without requiring synchronized ego-exo pairs. REAR adopts a dual-branch architecture: one branch extracts egocentric representations, while the other retrieves semantically relevant exocentric features. These are fused via a cross-view integration module that performs staged refinement and attention-based alignment. To mitigate class imbalance, a class-adaptive selector dynamically adjusts retrieval depth based on class frequency, and independent classifiers are trained with logit-adjusted cross-entropy. Extensive experiments across three benchmarks demonstrate that REAR achieves state-of-the-art performance, with significant gains in object recognition and tail-class accuracy. Code will be released upon acceptance.

## 1 INTRODUCTION

Egocentric video understanding has become a pivotal area in computer vision, emphasizing the analysis of first-person video to model human-environment interactions. In contrast to third-person videos, egocentric perspectives offer direct access to users' visual and motor activities, enabling a more granular understanding of object manipulations and social interactions. This unique vantage point underpins a range of real-world applications, including robotics (Kumar et al., 2020) and augmented reality (Li et al., 2024a;b). A core research task within this domain is Egocentric Action Recognition (EAR) (Li et al., 2015; Tan et al., 2023; Shiota et al., 2024), which focuses on classifying fine-grained verb and noun categories from continuous first-person video streams. EAR is fundamental to enabling intelligent assistive systems and context-aware wearable devices.

Existing approaches to EAR can be broadly categorized into several groups. Early methods adapt architectures from third-person video recognition (Sudhakaran et al., 2019; Wu et al., 2022; Chalk et al., 2024; Gowda et al., 2024; Zhang et al., 2025), but often underperform due to egocentric-specific challenges such as pronounced camera motion and viewpoint variability. Methods that incorporate egocentric-specific cues (Li et al., 2018; Wang et al., 2023c; Schoonbeek et al., 2024; Xu et al., 2023b; Shiota et al., 2024; Pei et al., 2025) show improved task alignment but frequently rely on additional sensors (e.g., eye trackers or inertial units), limiting scalability and generalization. More recent work explores large-scale egocentric video foundation models (Lin et al., 2022; Pramanick et al., 2023; Zhao et al., 2023; Wang et al., 2023a; Zhao & Krähenbühl, 2023) and their task-specific adaptations (Lyu et al., 2025; Wu et al., 2025; Xu et al., 2025a), which demonstrate promising transferability but remain constrained by limited dataset diversity and long-tailed category distributions. Another line of research seeks to leverage third-person video knowledge for EAR (Xu et al., 2023a; Xue & Grauman, 2023; Li et al., 2021; Truong & Luu, 2025), though they typically require temporally aligned ego-exo video pairs, which are expensive to collect at scale.

In this work, we propose Retrieval-augmented Egocentric Action Recognition (REAR), a novel framework that enhances egocentric video representations by leveraging external third-person (exo-

Figure 1: **An overview of five paradigms for Egocentric Action Recognition (EAR).** The figure shows the progression from (a) simple **Exo-to-Ego** transfer and (b) multimodal **Ego-Enhanced Transfer**, to (c) **Ego-Only** native training, (d) **Ego-Exo Paired** learning, and (e) **Exo-Augmented Ego** retrieval, highlighting their distinct architectures and recognition capabilities.

centric) knowledge without requiring large-scale pretraining or paired ego-exo datasets, as illustrated in Fig. 1. In contrast to prior work that either neglects egocentric-specific cues or depends on costly synchronized recordings, REAR effectively incorporates rich semantics from third-person videos while preserving egocentric characteristics. REAR operates in a dual-branch architecture. The target branch extracts egocentric features from the input video, while the retrieval-augmented branch identifies and encodes the most relevant third-person videos using a cross-view retrieval module. The two branches are fused via a cross-view integration module, producing a unified representation for action classification. Notably, verbs and nouns are predicted through separate classifiers, each optimized with Logit-Adjusted Cross-Entropy (LACE) to mitigate long-tail distribution bias.

To ensure strong egocentric recognition, we fine-tune a unified video encoder on the target egocentric dataset and share it across both branches. The retrieval module includes a cross-view retriever and a class-adaptive selector, which dynamically adjusts the number of retrieved videos per class. This reallocates retrieval resources from head to tail classes, improving representation learning for rare classes while controlling computational cost. The cross-view integration module refines exocentric features through staged feature fusion. First, a coarse aggregated representation is formed via multi-level similarity-weighted fusion. Then, an attention mechanism enables fine-grained ego-exo alignment, generating class-specific exocentric features tailored to the input video. These are fused with egocentric features to form the final, ego-dominant representation. Extensive experiments demonstrate that REAR achieves state-of-the-art results across three egocentric benchmarks. In particular, we observe substantial improvements in noun recognition for tail classes on EPIC-Kitchens-100, highlighting the effectiveness of retrieval-based augmentation under data scarcity.

The main contributions of our work are summarized as follows:

- We introduce **REAR**, a retrieval-augmented framework that enhances egocentric action recognition by incorporating external exocentric video representations without requiring paired datasets or additional sensors.

- We propose two core components: a **class-adaptive selector** that reallocates retrieval resources to tail classes, and a **cross-view integration module** that performs staged ego–exo feature fusion with high computational efficiency.

- REAR achieves strong performance across three egocentric benchmarks, with particularly notable improvements in noun recognition and long-tail categories on EPIC-Kitchens-100.

## 2 RELATED WORK

**Egocentric Action Recognition.** Existing approaches to EAR fall into four broad categories. The first line of work (Sudhakaran et al., 2019; Ashutosh et al., 2023; Gowda et al., 2024; Zhang et al., 2025) directly adapts third-person recognition models to egocentric scenarios. Although straightforward, they often fail to address egocentric-specific challenges, such as pronounced camera motion, viewpoint variation, and frequent hand-object occlusions. To address these limitations, subsequent works integrate egocentric-specific cues, including gaze (Li et al., 2018), head motion (Wang et al., 2023c; Schoonbeek et al., 2024), audio (Chalk et al., 2024), and hand-object interactions (Xu et al.,

2023b; Shiota et al., 2024; Pei et al., 2025). While these cues improve task alignment, they often require extra sensors or dense annotations, limiting scalability and real-world applicability.

Recent advances in Video Foundation Models (VFMs) have brought a new wave of research. Large-scale models such as EgoVLPv2 (Pramanick et al., 2023) and LaViLa (Zhao et al., 2023) learn general-purpose representations from egocentric data. Efficient adaptation techniques, including parameter-efficient fine-tuning (Wu et al., 2025), prompt learning (Lyu et al., 2025), and feature alignment (Xu et al., 2025a), further improve performance. However, the construction and deployment of such models remain resource-intensive and are limited by the class imbalance and scale of available datasets. Another promising direction is ego-exo joint learning (Ho et al., 2018; Xu et al., 2023a; Xue & Grauman, 2023), which utilizes synchronized multi-view data for improved representation learning. Yet, acquiring large-scale paired datasets (Sigurdsson et al., 2018; Kwon et al., 2021; Sener et al., 2022) is labor-intensive, and unsupervised alignment methods (Li et al., 2021; Wang et al., 2023b; Truong & Luu, 2025) introduce additional technical challenges.

While these strategies offer valuable insights, many rely on extensive resources or fail to fully exploit the unique characteristics of egocentric data. In contrast, we propose a retrieval-augmented framework that enriches egocentric representations by incorporating third-person video knowledge without requiring large-scale training, additional sensors, or paired datasets.

**Retrieval-Augmented Learning.** Retrieval-augmented learning emerged from memory-augmented neural networks (Graves et al., 2014) and gained traction in NLP through models that integrate retrieval with language generation (Guu et al., 2020; Lewis et al., 2020; Borgeaud et al., 2022), demonstrating the power of external memory for improving generalization and interpretability. In computer vision, retrieval-based augmentation has been applied to tasks such as open-world recognition (Liu et al., 2019), instance-level retrieval (Touvron et al., 2021), image generation (Chen et al., 2022b; Yasunaga et al., 2022), captioning (Sarto et al., 2022; Ramos et al., 2023; Xu et al., 2025b), and visual question answering (Chen et al., 2022a; Yang et al., 2022).

Closest to our work are retrieval-augmented methods for image recognition (Long et al., 2022; Iscen et al., 2023), which enhance classification by incorporating externally retrieved visual features. However, these approaches operate primarily in the image domain and typically rely on CLIP (Radford et al., 2021) for cross-modal retrieval, which is not well-suited to video understanding and struggles with cross-view alignment between egocentric and exocentric perspectives. To address this limitation, we adopt a cross-view retrieval module specifically designed for egocentric-exocentric video alignment (Xu et al., 2024), enabling semantically relevant third-person videos to be retrieved and integrated. This design facilitates egocentric action recognition without requiring time-synchronized multi-view recordings or additional supervision.

# 3 RETRIEVAL-AUGMENTED EGOCENTRIC ACTION RECOGNITION

## 3.1 OVERVIEW

As shown in Fig. 2, REAR consists of two parallel branches. The *target branch* extracts features from the input egocentric video, while the *retrieval branch* retrieves and encodes exocentric videos to provide complementary context.

Given an egocentric video $v^{\text{ego}}$, both branches share a unified visual encoder $E$, which ensures alignment in a common representation space. The target feature is computed as:

$$\boldsymbol{f}^{\text{ego}} = E(v^{\text{ego}}) \in \mathbb{R}^d. \tag{1}$$

Concurrently, the retrieval module identifies the $k$ most relevant exocentric videos:

$$\{v_1^{\text{exo}}, \ldots, v_k^{\text{exo}}\} = R(v^{\text{ego}}), \tag{2}$$

which are encoded using the same visual encoder:

$$\boldsymbol{f}_i^{\text{exo}} = E(v_i^{\text{exo}}) \in \mathbb{R}^d, \quad i = 1, \ldots, k. \tag{3}$$

The *Cross-view Integration Module* fuses the egocentric feature $\boldsymbol{f}^{\text{ego}}$ with the retrieved exocentric features $\{\boldsymbol{f}_i^{\text{exo}}\}_{i=1}^k$ to obtain an enhanced representation:

$$\boldsymbol{f} = \text{Integrate}\left(\boldsymbol{f}^{\text{ego}}, \{\boldsymbol{f}_i^{\text{exo}}\}_{i=1}^k\right) \in \mathbb{R}^d. \tag{4}$$

Finally, the integrated representation $\boldsymbol{f}$ is passed to parallel verb and noun classifiers, respectively.

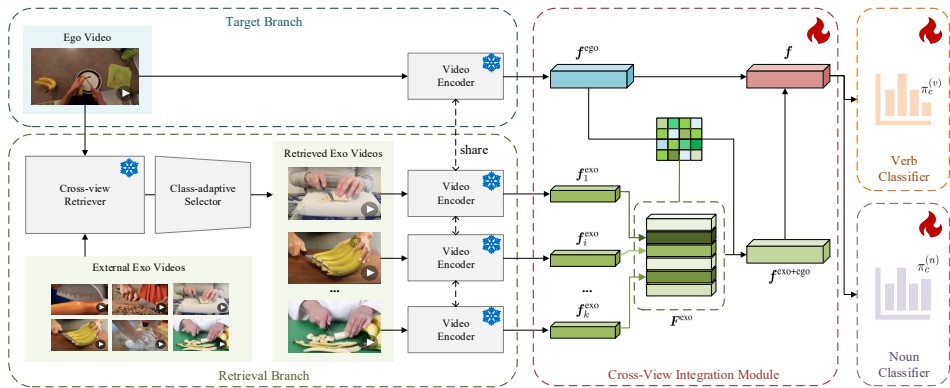

Figure 2: **An overview of REAR.** Given an egocentric video, the framework leverages relevant exocentric videos through retrieval and cross-view integration to enhance action recognition.

## 3.2 RETRIEVAL BRANCH

The retrieval branch comprises a *Cross-view Retriever* and a *Class-adaptive Selector*, jointly designed to retrieve relevant exocentric videos while accounting for the long-tailed class distribution.

### 3.2.1 CROSS-VIEW RETRIEVER

The goal of the Cross-view Retriever is to retrieve third-person (exocentric) videos that are semantically aligned with a given egocentric input. To this end, we adopt a pre-trained cross-view retrieval model (Xu et al., 2024) and fine-tune it on the target dataset using the EgoExoNCE loss, a contrastive objective tailored for cross-view video-text representation learning.

The model is trained with three types of positive pairs: (1) *intra-view video-text* (e.g., ego-ego and exo-exo), (2) *cross-view video-video* (e.g., ego-exo describing similar actions), and (3) *cross-view text-text* across views sharing at least one verb or noun. This strategy strengthens intra-view alignment and encourages robust cross-view generalization anchored by shared action semantics.

For a given egocentric video $v^{\text{ego}}$, we retrieve the top-$k$ exocentric videos from the candidate set $\mathbb{D}_v$ by computing an average of the video-video and video-text similarities:

$$\{v_1^{\text{exo}}, \dots, v_k^{\text{exo}}\} = \arg\max_k \left\{ \frac{1}{2} \left( \langle v^{\text{ego}}, v_i^{\text{exo}} \rangle + \langle v^{\text{ego}}, t_i^{\text{exo}} \rangle \right) \, \middle| \, v_i^{\text{exo}} \in \mathbb{D}_v, \, t_i^{\text{exo}} \in \mathbb{D}_t \right\}, \quad (5)$$

where $\mathbb{D}_t$ is the corresponding text descriptions, $t_i^{\text{exo}}$ is the caption associated with $v_i^{\text{exo}}$, and $k$ is dynamically determined by the Class-adaptive Selector (described next). This retrieval strategy ensures that selected exocentric videos are semantically relevant to the input while incorporating both visual and linguistic similarity cues.

### 3.2.2 CLASS-ADAPTIVE SELECTOR

To address the long-tailed class distribution, we introduce a Class-adaptive Selector that dynamically adjusts the number of retrieved exocentric videos $k$ based on the frequency of the predicted class. Concretely, we sort all action classes by their training frequencies and divide into three groups:

$$k(c) = \begin{cases} 20, & \text{if } c \in \text{tail (bottom 20\%)}, \\ 10, & \text{if } c \in \text{mid (middle 60\%)}, \\ 5, & \text{if } c \in \text{head (top 20\%)}. \end{cases} \quad (6)$$

This allocation strategy increases retrieval capacity for rare (tail) classes that lack sufficient egocentric training data, while limiting the overhead for well-represented (head) classes. Consequently, it provides a balanced trade-off between representational diversity and computational efficiency.

### 3.3 CROSS-VIEW INTEGRATION MODULE

The Cross-view Integration Module fuses information from egocentric and exocentric views in three stages: (i) similarity-guided feature aggregation, (ii) cross-view attention fusion, and (iii) final fea-

ture integration. This design enables robust representation learning while maintaining the discriminative power of egocentric features.

**Similarity-Guided Multi-Exo Feature Aggregation.** We first compute pairwise cosine similarity between the egocentric feature $\boldsymbol{f}^{\text{ego}}$ and each exocentric feature $\boldsymbol{f}_i^{\text{exo}}$:

$$s_i = \cos(\boldsymbol{f}^{\text{ego}}, \boldsymbol{f}_i^{\text{exo}}) = \frac{(\boldsymbol{f}^{\text{ego}})^\top \boldsymbol{f}_i^{\text{exo}}}{\|\boldsymbol{f}^{\text{ego}}\| \cdot \|\boldsymbol{f}_i^{\text{exo}}\|}, \quad i = 1, \ldots, k. \tag{7}$$

For higher-order relationships, all features are concatenated and fed into a two-layer ReLU MLP:

$$\boldsymbol{F} = [\boldsymbol{f}^{\text{ego}}; \boldsymbol{f}_1^{\text{exo}}; \ldots; \boldsymbol{f}_k^{\text{exo}}] \in \mathbb{R}^{(k+1) \times d}, \quad \boldsymbol{f}' = \text{MLP}_1(\boldsymbol{F}). \tag{8}$$

We then compute another similarity between the transformed feature $\boldsymbol{f}'$ and each exocentric feature:

$$s_i' = \cos(\boldsymbol{f}', \boldsymbol{f}_i^{\text{exo}}) = \frac{(\boldsymbol{f}')^\top \boldsymbol{f}_i^{\text{exo}}}{\|\boldsymbol{f}'\| \cdot \|\boldsymbol{f}_i^{\text{exo}}\|}. \tag{9}$$

The final attention weights are computed using a weighted softmax:

$$\alpha_i = \frac{\exp(w_1 s_i + w_2 s_i')}{\sum_{j=1}^k \exp(w_1 s_j + w_2 s_j')}, \quad i = 1, \ldots, k, \tag{10}$$

where $w_1$ and $w_2$ are learnable scalar weights. The resulting weighted exocentric features are:

$$\boldsymbol{F}^{\text{exo}} = [\alpha_1 \boldsymbol{f}_1^{\text{exo}}; \ldots; \alpha_k \boldsymbol{f}_k^{\text{exo}}] \in \mathbb{R}^{k \times d}. \tag{11}$$

**Cross-view Attention Fusion.** To enhance cross-view interactions, we adopt a query-key-value attention mechanism. The egocentric feature serves as the query, while re-weighted exocentric features provide the keys and values:

$$\boldsymbol{q} = \boldsymbol{f}^{\text{ego}} \boldsymbol{W}_{\text{q}}, \quad \boldsymbol{K} = \boldsymbol{F}^{\text{exo}} \boldsymbol{W}_{\text{k}}, \quad \boldsymbol{V} = \boldsymbol{F}^{\text{exo}} \boldsymbol{W}_{\text{v}}, \tag{12}$$

where $\boldsymbol{W}_{\text{q}}, \boldsymbol{W}_{\text{k}}, \boldsymbol{W}_{\text{v}} \in \mathbb{R}^{d \times d}$ are learnable projection matrices. The attended feature is then:

$$\boldsymbol{f}^{\text{exo+ego}} = \boldsymbol{A} \boldsymbol{V}, \quad \text{where } \boldsymbol{A} = \text{softmax}\left(\frac{\boldsymbol{q} \boldsymbol{K}^\top}{\sqrt{d}}\right). \tag{13}$$

**Final Feature Integration.** The final representation is obtained by concatenating the original egocentric and the attention-enhanced exocentric feature, followed by a non-linear transformation:

$$\boldsymbol{f} = \text{MLP}_2([\boldsymbol{f}^{\text{ego}}; \boldsymbol{f}^{\text{exo+ego}}]) \in \mathbb{R}^d. \tag{14}$$

### 3.4 LOSS FUNCTION

Given the fused representation $\boldsymbol{f}$, we apply two independent classifiers to predict verb and noun:

$$\boldsymbol{z}^v = C_v(\boldsymbol{f}) \in \mathbb{R}^{|\mathbb{V}|}, \quad \boldsymbol{z}^n = C_n(\boldsymbol{f}) \in \mathbb{R}^{|\mathbb{N}|}, \tag{15}$$

where $|\mathbb{V}|$ and $|\mathbb{N}|$ denote the number of verb and noun classes, respectively. To address class imbalance in egocentric datasets, we adopt the LACE loss (Menon et al., 2021), an effective and theoretically-grounded alternative to re-weighting that adjusts logits themselves based on class frequency. For each class $c$, a frequency-aware logit adjustment is applied:

$$\Delta_c = \tau \cdot \log(\pi_c), \tag{16}$$

where $\pi_c$ is the empirical class prior computed over the training set, and $\tau$ is a temperature hyper-parameter set to 1.0 based on validation performance. Separate priors are maintained for verbs and nouns: $\sum_{c=1}^{|\mathbb{V}|} \pi_c^{(v)} = 1$ and $\sum_{c=1}^{|\mathbb{N}|} \pi_c^{(n)} = 1$. The logit-adjusted cross-entropy losses for verb and noun predictions are:

$$\mathcal{L}_v = -\log \frac{\exp(z_{y_v}^v + \Delta_{y_v}^{(v)})}{\sum_{c=1}^{|\mathbb{V}|} \exp(z_c^v + \Delta_c^{(v)})}, \tag{17}$$

$$\mathcal{L}_n = -\log \frac{\exp(z_{y_n}^n + \Delta_{y_n}^{(n)})}{\sum_{c=1}^{|\mathbb{N}|} \exp(z_c^n + \Delta_c^{(n)})}, \tag{18}$$

where $y_v$ and $y_n$ denote the ground-truth labels for verb and noun classes. The total training objective is the sum of both losses:

$$\mathcal{L} = \mathcal{L}_v + \mathcal{L}_n. \tag{19}$$

# 4 EXPERIMENTS

We evaluate our REAR framework through comprehensive experiments. Section 4.1 details experimental setup. Section 4.2 reports main results and comparisons, while Section 4.3 and Section 4.4 present ablation studies and qualitative analyses, respectively.

## 4.1 EXPERIMENTAL SETTINGS

### 4.1.1 DATASETS

We evaluate REAR on three widely used egocentric video datasets, including EPIC-Kitchens-100, EGTEA Gaze+, and Charades-Ego, ensuring comparability with recent state-of-the-art methods. For external exocentric video retrieval, we use the YouCookII dataset.

**EPIC-Kitchens-100** (Damen et al., 2022) is the largest benchmark for egocentric action recognition, comprising 67K training clips and 10K validation/test clips. It captures kitchen activities and provides fine-grained annotations spanning 97 verb classes and 300 noun classes. Following Shiota et al. (2024), we analyzed the long-tailed distribution by ranking and dividing the label space into head (top 20%), middle (middle 60%), and tail (bottom 20%) splits based on sample frequency.

**EGTEA Gaze+** (Li et al., 2018) consists of 10,321 egocentric action clips annotated with 106 fine-grained action classes. The dataset is split into 8,299 training and 2,022 testing clips, with an average clip length of 3.2 seconds. All videos are recorded in kitchen environments, making this dataset complementary to EPIC-Kitchens-100 for evaluating kitchen-centric action recognition.

**Charades-Ego** (Sigurdsson et al., 2018) contains 68K video clips covering 157 action classes, recorded in diverse daily-life scenarios. Although the dataset includes time-synchronized egocentric and third-person views, we followed prior work (Sigurdsson et al., 2018) and used only the egocentric portion with the official train/test split for a fair comparison.

**YouCookII** (Zhou et al., 2018) is used as the external exocentric video pool. It consists of 2,000 untrimmed cooking videos sourced from YouTube, spanning 89 recipe classes and totaling 176 hours of video. Each video is temporally segmented and paired with natural language descriptions. We selected YouCookII as our retrieval corpus for three reasons: (1) its domain closely matches that of the egocentric datasets, (2) its rich textual annotations facilitate cross-modal and cross-view retrieval, and (3) its scale ensures wide coverage of relevant actions and objects.

### 4.1.2 BASELINES

We compare REAR against a comprehensive set of state-of-the-art methods published within the past three years. These baselines span four major categories:

- **Exocentric adaptation methods**, which transfer knowledge from conventional exocentric videos to egocentric domain. Representative methods include MeMViT (Wu et al., 2022), SFA-ViViT (Gowda et al., 2024), GACon (Zhang et al., 2025), and MACS (Lu et al., 2025).

- **Egocentric-specific approaches**, which exploit modality-specific cues such as gaze, hand-object interaction, or audio signals. This category includes EgoPCA (Xu et al., 2023b), HOCL-OSL (Shiota et al., 2024), and TIM (Chalk et al., 2024).

- **Video Foundation Models (Ego-VFMs)**, which leverage large-scale pretraining on egocentric datasets. This includes general-purpose models such as EgoVLPv2 (Pramanick et al., 2023), LaViLa (Zhao et al., 2023), Ego-Only (Wang et al., 2023a), and AVION (Zhao & Krähenbühl, 2023), as well as task-specific variants like Ego-VPA (Wu et al., 2025) and EgoPrompt (Lyu et al., 2025).

- **Ego-Exo joint learning methods**, which use multi-view video alignment to bridge egocentric and exocentric domains. Notable examples include SUM-L (Wang et al., 2023b), BYOV (Park et al., 2025), and CVAR (Truong & Luu, 2025).

In addition to these external baselines, we include a simplified variant, **REAR (baseline)**, which uses only the egocentric branch without retrieval-augmentation. All baseline results are cited directly from their original publications to ensure fair comparison under standard evaluation protocols.

Table 1: Comparison of our method with baselines on the Epic-Kitchens-100. The best and second-best results are **bold** and underlined, respectively.

| Method | Backbone | Verb (Tail) | Noun (Tail) | Action |
|---|---|---|---|---|
| MeMViT (Wu et al., 2022) | MViTv2 | 71.40 | 60.30 | 48.40 |
| AVION (Zhao & Krähenbühl, 2023) | ViT-L | 73.00 | 65.40 | 54.40 |
| EgoPCA (Xu et al., 2023b) | MViT | 68.70 | - | - |
| Ego-Only (Wang et al., 2023a) | ViT-L | 73.30 | 59.40 | - |
| LaViLa (Zhao et al., 2023) | TSF-L | 72.00 | 62.90 | 51.00 |
| SUM-L (Wang et al., 2023b) | SlowFast-R101 | 67.00 | 53.40 | - |
| HOCL-OSL (Shiota et al., 2024) | Swin-B | 54.33 (40.48) | 52.63 (33.72) | 33.52 |
| | SlowFast | 59.70 (38.13) | 45.16 (25.28) | 33.63 |
| SFA-ViViT (Gowda et al., 2024) | ViViT-L | 65.16 | 56.78 | 43.54 |
| TIM (Chalk et al., 2024) | Omnivore&VideoMAE-L | 76.20 | 66.40 | 56.40 |
| CVAR (Truong & Luu, 2025) | ViT-B | 69.37 (41.93) | 61.03 (38.58) | 46.15 |
| EgoPrompt (Lyu et al., 2025) | TSF-L | 61.40 | 44.58 | - |
| GACon (Zhang et al., 2025) | CLIP-400M | 69.50 | 58.10 | 45.70 |
| REAR (baseline) | UniFormerV2 | 71.70 (58.01) | 59.53 (35.01) | 53.50 |
| REAR (cross-view) | UniFormerV2 | **76.57** (67.35) | **67.80** (48.57) | **57.60** |

### 4.1.3 EVALUATION METRICS

We followed the standard evaluation protocols for each dataset to ensure fairness and consistency.

For **EPIC-Kitchens-100**, we report top-1 accuracy (%) for verb, noun, and action (verb+noun), as per the official benchmark. In addition, following prior work (Shiota et al., 2024; Truong & Luu, 2025), we report performance on tail classes to evaluate robustness under long-tailed distributions.

For **EGTEA Gaze+**, we report top-1 action recognition accuracy (%) across all test clips.

For **Charades-Ego**, we adopt mean Average Precision (mAP) as the evaluation metric, consistent with standard protocol (Sigurdsson et al., 2018).

### 4.1.4 IMPLEMENTATION DETAILS

We adopted UniFormerV2-L (Li et al., 2022) as the video encoder backbone and EgoInstructor (Xu et al., 2024) as the retrieval model backbone. Both components are first fine-tuned on the target egocentric dataset and then frozen during the main training phase of the REAR framework.

For video processing, we sample clips of 8 frames at a spatial resolution of $224 \times 224$, and extract 768-dimensional feature vectors using the frozen encoder. Text features, which are only used during training of the retrieval model, are extracted from BERT-base (Devlin et al., 2019) using action labels from the exocentric datasets and the sentence-level captions provided in the YouCookII corpus.

The cross-view integration module and classification heads are jointly optimized in an end-to-end manner. The integration module contains two MLPs and cross-attention projection matrices, each equipped with ReLU activations and Layer Normalization. We used the AdamW optimizer with an initial learning rate of $3 \times 10^{-5}$, weight decay of 0.01, cosine annealing learning rate schedule, and a batch size of 16. Models are trained for 30 epochs, with early stopping applied if validation accuracy does not improve for 5 consecutive epochs.

The temperature parameter $\tau$ in the LACE loss is set to 1.0, selected via validation performance. For the class-adaptive selector, we assigned $k$ based on class frequency: 20 for tail classes (bottom 20%), 10 for mid-frequency classes (middle 60%), and 5 for head classes (top 20%). All experiments are implemented in PyTorch and conducted on NVIDIA V100 GPUs.

## 4.2 PERFORMANCE COMPARISON

On **EPIC-Kitchens-100** (Table 1), **REAR** achieves state-of-the-art top-1 accuracy of 76.57% (verb), 67.80% (noun), and 57.60% (action), surpassing all prior methods. While the REAR (baseline) model underperforms compared to strong Ego-VFMs such as AVION (Zhao & Krähenbühl, 2023) and LaViLa (Zhao et al., 2023), the introduction of the retrieval-augmented branch yields substantial improvements of **+4.87%**, **+7.55%**, and **+4.10%** on verb, noun, and action, respectively. This confirms the effectiveness of the external knowledge to enhance egocentric feature representations. Compared with TIM (Chalk et al., 2024), which benefits from audio cues, REAR shows a mod-

Table 2: Comparison results on EGTEA.

| Method | Backbone | Action |
|---|---|---|
| EgoPCA (Xu et al., 2023b) | MViT | 70.80 |
| LaViLa (Zhao et al., 2023) | TSF-L | 71.37 |
| HOCL-OSL (Shiota et al., 2024) | Swin-B | 65.98 |
| | SlowFast | 66.86 |
| Ego-VPA (Wu et al., 2025) | TSF-L | 73.39 |
| MACS (Lu et al., 2025) | ViT | 67.30 |
| REAR (baseline) | UniFormerV2 | 69.07 |
| REAR (cross-view) | UniFormerV2 | **74.38** |

Table 3: Comparison results on Charades-Ego.

| Method | Backbone | mAP |
|---|---|---|
| EgoVLPv2 (Pramanick et al., 2023) | TSF-L | 34.10 |
| Ego-Only (Wang et al., 2023a) | ViT-L | 39.20 |
| LaViLa (Zhao et al., 2023) | TSF-L | 33.70 |
| SUM-L (Wang et al., 2023b) | SlowFast-R101 | 30.70 |
| BYOV (Park et al., 2025) | ViT-B | 31.80 |
| CVAR (Truong & Luu, 2025) | ViT-B | 31.95 |
| Ego-VPA (Wu et al., 2025) | TSF-L | 33.80 |
| REAR (baseline) | UniFormerV2 | 32.90 |
| REAR (cross-view) | UniFormerV2 | **40.70** |

Table 4: Ablation study on Epic-Kitchens-100 dataset. $\uparrow$ denotes improvement over baseline (#0).

| # | Top-$k$ | Integration | Loss | Verb Acc. | | | | | Noun Acc. | | | | |
|---|---|---|---|---|---|---|---|---|---|---|---|---|---|
| | | | | Head | Mid | Tail | All | $\uparrow$ | Head | Mid | Tail | All | $\uparrow$ |
| 0 | - | - | CE | 81.18 | 70.02 | 58.01 | 71.70 | - | 76.71 | 62.53 | 35.01 | 59.53 | - |
| 1 | 5 | SimAtt | CE | 82.11 | 71.08 | 61.87 | 73.58 | 1.88 | 77.31 | 63.05 | 39.87 | 61.93 | 2.40 |
| 2 | 5 | SimAtt | BalCE | 82.03 | 71.81 | 61.95 | 73.87 | 2.17 | 77.05 | 64.01 | 41.59 | 62.53 | 3.00 |
| 3 | 5 | SimAtt | Focal | 82.03 | 71.97 | 62.01 | 74.03 | 2.33 | 77.95 | 65.98 | 42.80 | 63.78 | 4.25 |
| 4 | 5 | SimAtt | LACE | 83.05 | 72.19 | 63.03 | 75.08 | 3.38 | 78.13 | 65.98 | 43.30 | 64.08 | 4.55 |
| 5 | 10 | SimAtt | LACE | 82.24 | 72.98 | 65.91 | 75.89 | 4.19 | 77.13 | 68.03 | 46.04 | 64.98 | 5.45 |
| 6 | 20 | SimAtt | LACE | 82.03 | 73.02 | 66.87 | 75.37 | 3.67 | 76.97 | 67.98 | 47.53 | 65.07 | 5.54 |
| 7 | Adp | Logit | LACE | 81.85 | 70.10 | 63.51 | 72.81 | 1.11 | 77.05 | 63.15 | 38.07 | 60.95 | 1.42 |
| 8 | Adp | AvePool | LACE | 81.54 | 71.14 | 63.87 | 73.03 | 1.33 | 76.93 | 63.17 | 38.79 | 61.03 | 1.50 |
| 9 | Adp | SimAg | LACE | 82.51 | 72.85 | 64.35 | 74.71 | 3.01 | 77.93 | 66.15 | 43.01 | 63.98 | 4.45 |
| 10 | Adp | CrossAtt | LACE | 82.57 | 72.53 | 65.38 | 74.83 | 3.13 | 78.03 | 66.94 | 43.32 | 64.32 | 4.79 |
| 11 | Adp | SimAtt | LACE | **83.33** | **74.38** | **67.35** | **76.57** | **4.87** | **78.86** | **68.75** | **48.57** | **67.80** | **8.27** |

est gain of **+0.37%** in verb recognition. However, REAR achieves notably higher gains in noun (**+1.40%**) and action (**+1.20%**), emphasizing its strength in noun recognition—a known bottleneck in egocentric action understanding due to higher visual ambiguity and occlusion. On **tail classes**, REAR (baseline) exhibits suboptimal noun recognition, lagging behind CVAR (Truong & Luu, 2025). Nevertheless, the retrieval augmentation improves tail-noun accuracy by **+13.56%**, ultimately outperforming both HOCL-OSL (Shiota et al., 2024) and CVAR (Truong & Luu, 2025) by **+14.58%** and **+9.99%**, respectively. These results highlight the efficacy of our class-adaptive retrieval strategy in enriching representations for underrepresented, information-scarce categories.

On **EGTEA** (Table 2), REAR achieves 74.38% top-1 action accuracy, and on **Charades-Ego** (Table 3), it attains 40.70% mAP. These consistent improvements demonstrate the generalizability of our framework across diverse egocentric benchmarks and validate the core premise of REAR: augmenting egocentric representations with external, semantically aligned exocentric knowledge leads to more robust and balanced action recognition, especially in long-tailed settings.

## 4.3 ABLATION STUDY

We conducted detailed ablation experiments on EPIC-Kitchens-100 (Table 4) to evaluate the contributions of each component. The following abbreviations are: **Top-$k$** denotes the number of retrieved exocentric videos; **Adp** refers to the class-adaptive selector for dynamic $k$ adjustment; **Logit** indicates logit-level fusion (Long et al., 2022); **SimAtt** denotes our staged fusion module, which combines Similarity-guided multi-exo Aggregation (**SimAg**) with Cross-view Attention (**CrossAtt**).

Our full model (#11) consistently outperforms the baseline (#0), achieving relative gains of **+4.87%** in verb recognition (**+9.34%** for tail classes) and **+8.23%** in noun recognition (**+13.56%** for tail classes). As shown in the class-wise analysis (Fig. 3), the retrieval-augmented branch leads to especially notable improvements in noun recognition and performance on underrepresented categories.

**Effect of top-$k$ selection.** Comparing #0 and #4-#6, we observed that head classes (top 20%) benefit minimally from additional retrieved data. Mid-frequency classes show modest gains when increasing $k$ from 10 to 20, while tail classes benefit substantially from $k = 20$. These trends support our adaptive retrieval strategy, which improves performance by allocating more external knowledge to underrepresented classes while maintaining computational efficiency for frequent ones.

**Effect of integration strategies.** Results from #7-#11 reveal that both direct logit-level fusion (Long et al., 2022) and naïve average pooling (#8) underutilize retrieved exocentric information,

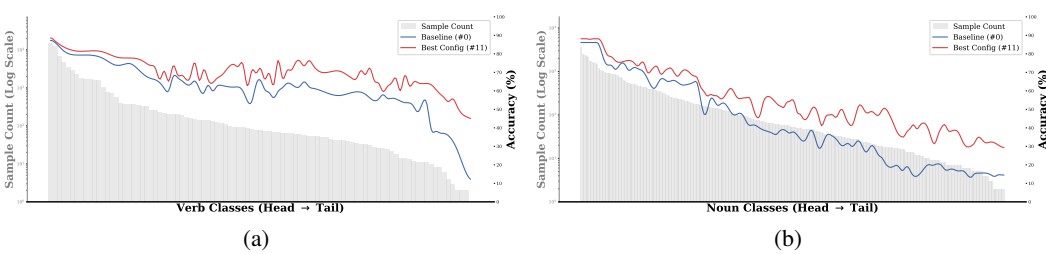

(a)                                                      (b)

Figure 3: **Class-wise performance and distribution for (a) verbs and (b) nouns.** Bars show counts; lines compare Baseline (#0) vs Best Config (#11), highlighting stronger gains for tail classes.

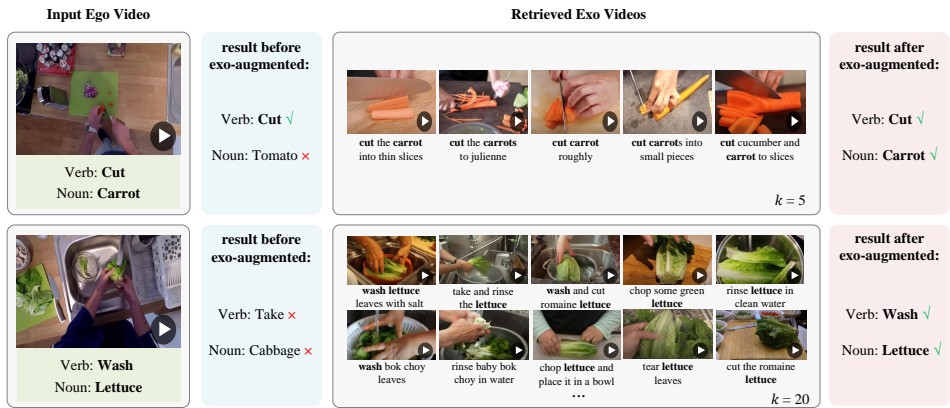

Figure 4: **Qualitative results.** Retrieved exocentric videos enhance egocentric action recognition, improving fine-grained object and action distinction.

due to the lack of explicit cross-view interaction. Incorporating either similarity-guided weighting (#9) or attention-based fusion (#10) improves performance. Combining both via our staged fusion module (#11) yields the strongest results, demonstrating that coarse-to-fine fusion effectively captures both global relevance and fine-grained alignment across views.

**Effect of loss functions.** In comparison of loss functions (#1-#4), we evaluated methods such as Balanced Softmax Cross Entropy (BalCE) (Long et al., 2022) and Focal loss (Lin et al., 2017). The results show that the LACE loss (#4) significantly improves tail-class accuracy by **+1.16%** (verbs) and **+3.34%** (nouns) over standard cross-entropy (#1). This underscores the importance of incorporating class priors to mitigate long-tail bias in egocentric action recognition.

### 4.4 QUALITATIVE RESULTS

As shown in Fig. 4, we present two sets of exocentric videos retrieved for given egocentric inputs. The examples illustrate that the retrieved samples generally correspond to the actions or objects in the egocentric video. By leveraging these exocentric videos, the model can more accurately recognize fine-grained verbs and nouns. In contrast, without this external enhancement, the model may misclassify "lettuce" as the more familiar, head-class "cabbage", with more training samples, and struggle to distinguish actions like "wash" and "take". These results demonstrate the effectiveness of REAR in utilizing retrieved third-person videos to enhance egocentric action recognition.

## 5 CONCLUSION

In this work, we introduce **REAR**, a retrieval-augmented framework for egocentric action recognition that leverages external third-person (exocentric) videos to enhance egocentric representations. Unlike prior methods that rely on paired ego-exo data or overlook egocentric-specific challenges, REAR integrates a class-adaptive retrieval mechanism with a staged cross-view fusion module to enable effective knowledge transfer without requiring additional sensors or alignment. Extensive experiments on three benchmarks demonstrate the effectiveness and generalizability of our approach.

## A  ETHICS STATEMENT

We have carefully considered the ethical implications of our research in accordance with the ICLR Code of Ethics. Our work focuses on fundamental machine learning principles and is restricted to standard academic benchmarks. This research did not involve any human subjects or participants.

The datasets used in this study are publicly accessible and have been widely used in previous research. They do not contain sensitive or private information. Specifically:

- **EPIC-Kitchens-100** (Damen et al., 2022) is used in accordance with its Creative Commons Attribution-NonCommercial 4.0 International (CC BY-NC 4.0) license.
- **EGTEA Gaze+** (Li et al., 2018) is used under the non-commercial research license specified on its official project website by its creators at Georgia Tech.
- **Charades-Ego** (Sigurdsson et al., 2018) is used under the academic research license provided by the Allen Institute for Artificial Intelligence.
- **YouCookII** (Zhou et al., 2018) dataset's annotations are provided under the MIT License, while the raw video files are distributed for non-commercial, research purposes only, a condition we strictly adhere to.

## B  REPRODUCIBILITY STATEMENT

To ensure the reproducibility of our research, we commit to making our source code publicly available upon publication of this paper. All implementation details, hyperparameters, and computing environment required to reproduce our experiments are detailed in Section 4.1.4. The datasets used are all publicly available and are cited in the main text.

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
