# OpenReview forum: "REAR: Retrieval-Augmented Egocentric Action Recognition"
_ICLR.cc/2026/Conference — ICLR 2026 Conference Withdrawn Submission_

### Official Review · Reviewer_ep8L · 2025-10-31

**Soundness:** 2
**Presentation:** 2
**Contribution:** 1
**Rating:** 2
**Confidence:** 4

**Summary:**

REAR is a retrieval-augmented egocentric action recognition framework that enriches ego features with top‑k exocentric videos retrieved via an EgoExoNCE‑trained cross‑view retriever. A dual‑branch shared encoder fuses retrieved exo features through similarity‑weighted aggregation and cross‑view attention, while a class‑adaptive selector increases k for tail classes and LACE aids long‑tail training. Across EPIC‑Kitchens‑100, EGTEA, and Charades‑Ego, REAR delivers consistent gains—especially on tail classes.

**Strengths:**

- Performance is strong across benchmarks, with consistent gains.
- Particularly effective for long-tailed and noun recognition, a known bottleneck in egocentric understanding.

**Weaknesses:**

- Academic novelty relative to EgoInstructor [1] is unclear. The core concepts—cross-view ego↔exo retrieval trained with EgoExoNCE and augmenting/fusing them for better downstream task—are inherited. REAR primarily swaps the downstream task from caption generation to closed-set action recognition and adds some engineering and modest architectural modification. Demonstrating transfer of retrieval augmentation to recognition is useful, but the conceptual advance beyond EgoInstructor appears incremental.

- Closest related work is mispositioned. The manuscript identifies retrieval-augmented image recognition as the closest prior, but EgoInstructor (Retrieval-Augmented Egocentric Video Captioning) is the most directly related and should be foregrounded with a clear distinction of what is genuinely new.

- The class-adaptive selector is heuristic.


[1]: Xu, Jilan, et al. "Retrieval-augmented egocentric video captioning." Proceedings of the IEEE/CVF Conference on Computer Vision and Pattern Recognition. 2024.

**Questions:**

The previous work, Egoinstructor, addresses the more challenging task of captioning, whereas this work focuses on closed-set action recognition, which is arguably an easier problem. I'm not convinced that this shift to a simpler task, on its own, constitutes a meaningful or novel contribution.

---

### Official Review · Reviewer_zKTe · 2025-10-31

**Soundness:** 2
**Presentation:** 3
**Contribution:** 2
**Rating:** 6
**Confidence:** 4

**Summary:**

The paper presents a two-stream architecture that implements a RAG framework for egocentric video action recognition. The contribution are on the retrieval mechanism and overall model design. Experiments show significant gains with including the RAG branch, and achieves sota results over recent baselines in four datasets.

**Strengths:**

This is one of the first works on RAG for video action recognition combining ego and exo view, which is an interesting research direction also for zero-shot. The model design is relatively simple and intuitive, and the components seem appropriately chosen and adapted to realize the framework. The paper is well written.

**Weaknesses:**

The RAG integration and overall architecture design is effective but not particularly original in its components. It is combining cross-attention, softmax for weighting, and LACE loss adjustment.

Equation 6 states that there are 3 different values for k while the MLP in 8 uses k stacked features as input. It is not detailed how the MLP deals with three different input lengths - using three MLPs with some weights sharing mechanism? Please clarify (see Questions section below).

There is no ablation or motivation on including $f^{\text{exo}}_i$ also in $f'$ in Eq.10. Please clarify (see Questions section below).

Resources needed for training and inference are not specified. Please clarify (see Questions section below).

**Questions:**

Please clarify how the MLP in eq.8 deal with the three different input lengths - using three MLPs with some weights sharing mechanism?

Provide an ablation and motivation on including $f^{\text{exo}}_i$ also in $f'$ in Eq.10.

Provide details on the resources needed for training and inference are not specified. It is stated that "conducted on NVIDIA V100 GPUs" - how many? For how long? What is inference latency?

---

### Official Review · Reviewer_komg · 2025-10-31

**Soundness:** 1
**Presentation:** 3
**Contribution:** 3
**Rating:** 2
**Confidence:** 5

**Summary:**

This paper focuses on the task of egocentric action recognition in which exocentric information is used in a retrieval augmented strategy. The paper proposes a new method which first finds the closest exocentric videos from a corpus for an inputted egocentric video (which represents a different dataset). These closest exocentric videos are then combined to help inference for the inputted egocentric video for the relevant actions/verb/noun classification. The method is tested across three egocentric datasets (EPIC-Kitchens, EGTEA Gaze+, Charades Ego) and YouCook2 as the exocentric gallery of videos. The results show that the method achieves the new state of the art results across all datasets.

**Strengths:**

* The idea of using retrieval from the exocentric domain is a nice concept and is implemented in a way that is not over-complicated, makes sense, and isn't a naive fusion method.
* The motivation of the method and the paper is strong and well formulated.
* The paper writing is generally clear and easy to follow.
* Results show the method's performance across the long-tail of actions with the method performing well on the tail classes.

**Weaknesses:**

* $R$ is not specified in equation 2, I assume this is the retrieval module, but this does not use the visual encoder? Reading onwards it becomes a bit more clear that the cross-view retriever is different, but why is a separate model utilised in the cross-view retriever instead of the same video encoder which puts the videos into the same space anyway?
* In the case in which the tail classes have less than 20 examples, are all examples still chosen? Given the size of the test set and the imbalanced nature of the dataset, is this a worry when performing inference?
* It is hard to compare the results within Table 1 as all the models use different backbones and are thus incomparable. In this way, it is impossible to know whether REAR achieves state of the art performance because of the backbone of UniFormerV2 or because of the proposed aspects of the method. Including experiments with a ViT-L backbone which gives the strongest previous performance for Ego-Only and AVION or re-training AVION/Ego-Only with a UniFormerV2 backbone would help clarify this. Unfortunately, because of this, the results are inconclusive even with the positive increase in performance of REAR from baseline to cross-view.
* The similarity calculation which uses the average of the video-video and video-text similarity calculation is defined but not ablated, it would be good to know how important this decision choice in the model's performance
* Missing citation for Ego-Exo4D [1] as a dataset which includes both first person and third person videos within the related work
* It would be good to know the impact of the video gallery, there are open questions regarding:
  * How does the size of the gallery impact the method's performance?
  * If an egocentric gallery was used, does this have a big impact on the method's performance, does this allow for better performance as the modalities are closer? Or is this worse because there is less complementary information from the exocentric perspective.
  * How relevant do the videos within the gallery need to be? For example, if an exocentric dataset which has less relevance was used does this harm performance by a large amount?


[1] Grauman, Kristen, et al. "Ego-exo4d: Understanding skilled human activity from first-and third-person perspectives." Proceedings of the IEEE/CVF Conference on Computer Vision and Pattern Recognition. 2024.

**Questions:**

1. Have fair comparisons between models with the same underlying backbone been investigated?
2. Has an exploration into the gallery and how this affects the model performance been completed?
3. Why is a different model used to find the relevant models in the exocentric modality?
4. Was the average video-video and video-text similarity calculation ablated?
5. In the case in which the tail classes have less than 20 examples, are all examples still chosen?

---

### Official Review · Reviewer_kvqx · 2025-11-01

**Soundness:** 4
**Presentation:** 4
**Contribution:** 4
**Rating:** 6
**Confidence:** 4

**Summary:**

This paper addresses the critical problem of long-tail class distributions and data scarcity in Egocentric Action Recognition (EAR). The authors propose REAR, a novel retrieval-augmented framework that enhances egocentric video representations by leveraging knowledge from abundant, external third-person (exocentric) videos. A key innovation is that this approach does not require strictly paired or synchronized ego-exo data, making it more practical and scalable. Experiments on three benchmarks demonstrate that REAR achieves state-of-the-art performance, showing significant improvements, particularly in recognizing actions and objects belonging to the long tail.

**Strengths:**

1. The paper is supported by a rigorous and well-designed experimental evaluation. The authors validate their proposed REAR framework across multiple standard egocentric action recognition benchmarks. The choice of baselines is comprehensive, covering a range of existing paradigms and ensuring a fair comparison.

2. The authors creatively leverage retrieval-augmented knowledge from external, data-abundant exocentric videos to bolster the performance on rare, tail classes. This presents an inspiring and effective strategy for knowledge transfer that directly targets the data scarcity issue at the heart of the long-tail problem.

3. The paper includes a set of ablation studies that systematically dissect the REAR framework and validate the contribution of each proposed component, to both overall performance and challenging tail classes.

**Weaknesses:**

1. No resource-usage information is mentioned in the paper, which is also an important aspect for evaluating the practical applicability of the proposed method. I would suggest the authors to include the following details:

    a) How many GPUs are used for training?

    b) How does the training time compare with baseline methods?

    c) Computation overhead comparison: the number of parameters, FLOPs.

    d) Memory usage during both training and inference.

    e) How does inference latency compare with baseline methods?

2. In Table 4, it lacks the loss comparison under the 'Adp + SimAtt' setting. Will the LACE loss still excel at mitigating class imbalance problem under that setting?

**Questions:**

See Weaknesses.

---

### Note · Authors · 2025-11-13

I have read and agree with the venue's withdrawal policy on behalf of myself and my co-authors.